# Do We All Live Story-Shaped Lives? Narrative Identity, Episodic Life, and Religious Experience

## Eunil David Cho

Brite Divinity School, Texas Christian University, Fort Worth, TX 76109, USA; e.cho@tcu.edu

**Abstract:** This article focuses on exploring the concept of narrative identity, which has emerged as an integrative concept in various academic fields. Particularly in philosophy and psychology, scholars have claimed that humans are storytellers by nature and tell their stories that develop in them a sense of identity. However, this concept has been criticized by those who have argued that while some people are Diachronic (narrative), some are Episodic (non-narrative). People with an episodic disposition do not or are not able to live a narrative or story of some sort. In order to explore the distinction between Diachronic and Episodic dispositions, I analyze the autobiographical writing of Leo Tolstoy, namely Tolstoy's personal religious experience presented in William James' "The Varieties of Religious Experience". This particular case study demonstrates how an Episodic person can become Diachronic and gain a sense of unity and a sense of self through religious experience. In the end, I argue that Episodic and Diachronic dispositions are not mutually exclusive in an individual's life, but that individuals may at different points in life experience their lives in one manner or another.

**Keywords:** narrative identity; life story; autobiography; episodic memory; virtue ethics; narrative unity; storytelling; episodic disposition; non-narrative; religious experience; religious identity





## 1. Introduction

Narrative identity refers to the sense of self one forms through the stories one constructs in order to understand who one is and to communicate this sense of self to others. Beginning in adolescence and young adulthood, narrative identities are often understood to be formed through the lived, internalized, and evolving stories by which humans live. The idea that human identity is created through stories, both consciously constructed and those absorbed through the family and culture, has emerged as an integrative concept in academic fields as diverse as philosophy, psychology, theology, anthropology, sociology, political theory, religious studies, psychotherapy, and medicine. For some time now, scholars in theology and religious stories have used narrative identity theories to explain how humans tell and live their stories in close interaction with the stories of a religious tradition (Ganzevoort 2012). In this article, I begin by examining closely how scholars in philosophy and psychology have investigated the use of stories to tell us "who we are" and provide us with our sense of self. In particular, I focus on reviewing Alasdair MacIntyre's contribution to the ethical perspective on narrative identity formation and Dan P. McAdams' seminal work on psychological narrative identity development.

While the idea of a story as a human identity has been widely accepted, it has also received some fierce criticism. Galen Strawson is one of the leading voices in anti-narrative discourses who highlights the "Episodic" mode of experiencing life, which means some people do not and are not able to experience their lives as a narrative or story of some sort. How are "Episodic" modes (non-narrative) and "Diachronic" (narrative) modes of self-experience different? How can people with the "Episodic" disposition develop the narrative mode of living? Is it possible for the Episodics to develop coherent narrative identities? To consider these questions, I examine the religious autobiography of Leo

Tolstoy presented by William James in "The Varieties of Religious Experience", which tells a specific story of how Tolstoy, with suicidal melancholia, gains a sense of coherence and clarity in his episodic state of life by having religious experiences. The analysis of the case study demonstrates how religion provides a resource for Episodics to begin to develop the narrative mode of experiencing life and find a sense of identity in their stories. Specifically, James' study of religious experience in "The Varieties" claims that religious experience as a psychological process is "the process of unification" that can bring unity and coherence to the "Episodic" mode of experiencing life.

## 2. Understanding the Ethical and Psychological Narrativity Theses

In his thought-provoking essay, "Against Narrativity," Strawson mainly criticizes the idea that many scholars in various fields often hold: To be human is to live a story. Strawson first focuses on the concept of narrativity to explain how people narrativize their stories and identities, but does not give a specific definition. A clearer definition of narrativity that is closely aligned with Strawson's understanding, is provided by John Christman: "an identifiable characteristic of the sequence of memories, reflections, actions, mental events or other such factors that mark them out as unified and individualized" (Christman 2004, p. 696). Narrativity is what makes one's story coherent, unified, and tellable. Strawson begins by introducing two narrative identity theses contained in the concept of narrativity that have been widely accepted: (1) the ethical, and (2) the psychological. First, the ethical narrativity thesis argues that experiencing or conceiving one's life as a narrative shapes a good life, providing a moral framework for a well-lived life to true or full personhood (Strawson 2004, p. 428). Second, the psychological narrativity thesis claims that "human beings typically see or live or experience their lives as a narrative or story of some sort, or at least as a collection of stories" (Strawson 2004, p. 428). Before the examination of Strawson's critique on these renowned narrativity theses, in the following section, I first evaluate MacIntyre's contribution to the ethical dimension to narrative identity in critical conversation with his feminist critic, Lisa Tessman. The subsequent section then explores the development of the psychological narrative identity thesis in personality psychology and developmental psychology based on the work of McAdams.[1]

### 2.1. The Ethical Narrativty Thesis: Achieving Narrative Unity to Live a Virtuous Life

Alasdair MacIntyre claims that the self is the product of a narrative, one's life story. In his notable work, "After Virtue", he explains: "narrative history of a certain kind turns out to be the basic and essential genre for the characterization of human action" (MaIntyre 2012, p. 208). Humans live out narratives in their lives, and precisely because they understand their own lives in terms of stories or narratives, the form of a narrative is appropriate for understanding the actions of humans (MaIntyre 2012, p. 212). Humans identify "who they are" by authoring and telling their own stories. MacIntyre expands the concept of identity by linking the notion of virtue with self-identity, claiming that virtues are dispositions and skills that allow individuals to carry intentions into action, to deal with circumstances in their lives, and to make and execute plans with purpose. The same virtues, MacIntyre argues, have to be equally and continuously manifested in "very different types of situations" (MaIntyre 2012, p. 205).

By underlining the importance of the continuity of virtues throughout one's life, MacIntyre introduces the idea of "the unity of an individual's life," in which events are connected in one coherent narrative sequence:

> And the unity of a virtue in someone's life is intelligible only as a characteristic of a unitary life, a life that can be conceived and evaluated as a whole . . . it has been necessary to say something of the concomitant concept of selfhood, a concept of

---

[1]　I focus on analyzing the ethical narrativity thesis of MacIntyre and the psychological narrativity thesis of McAdams because these two are among the narrativist scholars that Strawson argues against in his anti-narrative discourse.

a self whose unity resides in the unity of a narrative which links birth to life to death as narrative beginning to middle to end. (MaIntyre 2012, p. 205)

Moreover, a unitary life in the form of narrative acquires meaning, MacIntyre explains, only in the context of a tradition, because every person is the bearer of a tradition that is "historically extended" and "socially embodied ... through many generations" (MaIntyre 2012, p. 222). The tradition serves as a moral ledger that shapes and determines which acts, intentions, and individuals are virtuous. Thus, for MacIntyre, the virtuous self is the one who is able to narrate one's life story intelligibly in a meaningful way, aligned profoundly with a particular tradition or at least with the resources of the tradition.

Furthermore, MacIntyre's notion of narrative identity has a teleological character, claiming that the narrative unity of a life is lived and constructed around an ultimate goal (MaIntyre 2012, p. 216). Human life can be understood to have the structure of narrative with a beginning, middle, and end, united by a central, purpose-giving goal like "a quest" or "a journey" (MaIntyre 2012, p. 205). Such a goal gives not only a certain direction in life, but also meaning to a person's individual actions by connecting diverse and scattered events. For MacIntyre, to know oneself is "to find oneself placed at a certain point on a journey with set goals; to move through life is to make progress—or to fail to make progress—toward a given end" (MaIntyre 2012, p. 205). This human tendency toward the ultimate goal or intended purpose is in accordance with the Aristotelian concept of natural teleology, where one's life, even though multifaceted, can be perceived as a united structure moving toward the ultimate goal or telos. Therefore, in order to live a completed, fulfilled, and virtuous life, MacIntyre argues, one must achieve and live out "the narrative unity of life" with a clear sense of telos.

MacIntyre's notion of the narrative unity of life has contributed to the rise of narrative identity theories, namely the ethical narrative identity thesis, by claiming, "narrative history of a certain kind turns out to be the basic and essential genre for the characterization of human actions" (MaIntyre 2012, p. 208). A personal life story is continuously embedded in and connected to the historical narratives of the communities to which humans belong. Yet, MacIntyre's idea of identity as a narrative or story has been canonized enough to receive some criticism. In particular, many scholars question and challenge his idea of "the narrative unity" in a single life. These critics raise questions about how we could possibly keep and make sense of the stories of a community that may be subordinate or inferior. For those who might have been marginalized and disadvantaged by harmful historical narratives of their particular traditions, how are they to find a sense of unity in life and construct narratives with respect to their community's traditions?

One critic who has raised such questions is Lisa Tessman, a feminist virtue ethicist. Tessman claims that marginalized communities often encounter "the systemic barriers that make it difficult or impossible for those who are oppressed to gain or be granted the 'external' goods, including freedom, material resources, political power, and respect or social recognition of personhood—that are needed to live well" (Tessman 2001, p. 80). These individuals might not have been given equal opportunities to tell and develop their own stories of virtue. They also might not be able to see themselves in the stories of their oppressed or subjugated cultures. Oppression interferes with fundamental human flourishing, Tessman suggests, not only through the denial of goods, but also by structurally diminishing the possibilities for the development of virtue (Tessman 2001, pp. 79, 80). According to Aristotelian ethics, when one is prevented from flourishing, one can fail to fully develop and exercise virtues; one's own character can stand in the way of the good and ethical life (Tessman 2001, p. 80). Tessman characterizes these oppressed individuals who lack virtue as "morally damaged," implying that they would not be able to live a "virtuous life" and achieve "the narrative unity of life" (Tessman 2005, p. 333). For these compelling reasons, Tessman evaluates MacIntyre's concepts as a limiting set of principles and values, one which may privilege certain kinds of experiences over others.

Despite its limitations, MacIntyre's discourse on narrative identity provides the valuable insight that the construction of human identity revolves around the question of "what

is a good life?" adopting a structured narration informed by traditions as a basis for the formation of personal identity (Mela 2011, p. 103). MacIntyre maintains that there is a particular "narrative unity of human life," which is the conception of life as a "quest" or "journey." While MacIntyre's ethical narrativity thesis claims that in order to live a virtuous life, people need to find a sense of unity in their lives by telling stories, the psychological thesis focuses more on exploring people as storytellers by nature and how they make sense of their sense of identity, existence, purpose, and meaning by telling stories within and from their own contexts.

### 2.2. The Psychological Narrativity Thesis: We Are the Stories We Tell

Human beings are storytellers by nature. In a multitude of guises, as folktale, legend, myth, fairy tale, history, epic, opera, motion picture, television situation comedy, novel, biography, joke, and personal anecdotes, the story appears in every known human culture. We expect much from stories. We expect them to entertain, educate, inspire, and persuade us; to keep us awake and put us to sleep; to make us feel joy, sadness, anger, excitement, horror, shame, guilt, and virtually any other emotion we can name (McAdams 2013, p. 55).

When we want to communicate something about ourselves to others, we do this simply by telling a story. In fact, everyday conversation among people is storytelling of one form or another. Indeed, many scholars in the psychological sciences have suggested that the human mind is first and foremost a vehicle for storytelling (Howard 1989). As humans tell stories, we expect our stories to communicate with those to whom we tell them. We also expect stories to reveal our sense of identity. Dan P. McAdams, one of the leading psychologists who developed the theory of narrative identity from the specific perspective of personality psychology as well as developmental psychology, argues that "when it comes to human lives, storytelling is sense-making" (McAdams 2013, p. 55; McAdams 1997). McAdams starts with the eminent psychologist Jerome Bruner, who is known as one of the pioneering scholars on the use of stories in human lives. Bruner introduces two primarily different forms of human knowing: paradigmatic knowing and narrative knowing (Bruner 1986). Paradigmatic knowing is "the knowing of cause and effect, scientific and rational discourse" (McAdams 2013, p. 56). Its primary aim in knowing is to find "the single, logical, causal truth" (McAdams 2013, p. 56). In contrast, Bruner claims that narrative knowing is what people learn from stories that, in essence, "convey and explain human conduct" (McAdams 2013, p. 56). For Bruner, stories are basically about "the vicissitudes of human intention" organized in time (Bruner 1990, p. 17). In other words, stories are about "what characters want, what they intend to do, and how, over time, they go about trying to get what they want or avoid what they don't want" (McAdams 2013, p. 56). Thus, by telling stories, people come to understand each other's motivations, intentions, aspirations, needs, and ideas—they can come together more efficiently in matters of communal concern. So much of human adaptation involves social cooperation, McAdams explains that "storytelling may promote fitness—for the individuals and the group—by making more transparent individual intentions and helping to coordinate different people's wants and needs in order to accomplish such broad social goals" (McAdams 2013, p. 58). That is why human beings have evolved—to engage in storytelling in order to understand various tales of human experience to be profoundly captivating and meaningful. Sharing stories with and hearing stories from one another provides a space for people to enrich their interpersonal relationships and cultivate a sense of community.

As McAdams explains, stories that are based on personal experiences begin with episodic memory. Episodic memory is "the ability to recall specific events (episodes) from the past" (McAdams 2013, p. 58). As the memory of autobiographical data, it is a skill that enables humans "to travel backward in subjective time and to link remembered events to imagined future episodes" (McAdams 2013, p. 58; Tulving 2002). Episodic memory provides the personal experience of time that people depend upon when they construct and tell stories. A considerable amount of what people remember may not be directly connected in their minds to specific occasions and events. Nonetheless, human beings are still able to travel back in time to recall specific episodes that have happened in their

lives. McAdams contends that episodic memory provides people with the feeling that their lives are set in an ongoing timeframe that includes "the remembered past, the experienced present and the anticipated future" (McAdams 2013, pp. 59, 60). The main function of episodic memory is to provide the foundation for identity formation—people's narrative sense of self-in-time.

How, then, is narrative identity created? How is human identity shaped and developed through narrating life's story? Humans start putting their lives together into personalized life stories in their adolescent and young adult years (McAdams 2013, pp. 62, 63). The notion of timing corresponds agreeably to the human life course of Erik Erikson, who understood adolescence as the period of identity formation (Erikson 1959). Usually, by the time people reach their teen years, they are expected to have thought about the following questions: Who am I? What do I want to do when I grow up? What do I truly believe in? Where is the purpose of my life? What is the meaning of my life? According to McAdams, people begin asking these kinds of questions about life—identity questions—in the years of "emerging adulthood" (McAdams 2013, p. 62). People consciously and unconsciously ask these questions because, in this specific time period in life, the cultural conditions and expectations of modern society are such that "you are supposed to ask these questions of yourself." Before the time of young adulthood, people begin telling their stories. However, it is not until the period of emerging adulthood that people begin to select and arrange their entire lives—"the past as they remember it, the present as they perceive it, and the future as they imagine it"—into a life narrative by using episodic memories (McAdams 2013, p. 62). Thus, McAdams defines the narrative self as "the internalized and changing story of your life that people begin to work on in the emerging adult years" (McAdams 2013, p. 62).[2] The story as identity brings together into a narrative the many different ideas, values, and hopes people have and the roles they play, and this narrative serves as "a flexible guide for the future and an historical archive for making sense of your past" (McAdams 2013, p. 63). In other words, narrative identity becomes shaped and structured as we remember and comprehend our past memories, present situations, and future aspirations, weaving them into a narrative form. As such, as they make and tell stories, human beings create their own identities: they have agency to decide who they are, who they were, and who they may become.[3]

### 3. Against Narrativity: Does Everyone Have a Story-Shaped Life?

In his argument against the two narrative identity theses that have been widely accepted in academic fields as diverse as "philosophy, psychology, theology, anthropology, sociology, political theory, literary studies, religious studies, psychotherapy, and even medicine," Strawson explains that these two perspectives are not logically linked (Strawson 2004, p. 428). People may hold to the psychological thesis (we see our lives as a narrative), but not the ethical thesis (we do not need to live out a narrative in order to live a good life). Likewise, people may cling to the ethical thesis (we should live a storied life), but not the psychological thesis (but people typically do not see their lives as a narrative).

#### 3.1. Diachronics vs. Episodics

Based on his observation, Strawson raises the subsequent question: does every human being live a story-shaped life? In response, he then proceeds to divide humanity into two ways of being: (1) Diachronics and (2) Episodics. Diachronics are those who see themselves "as something that was there in the (further) past and will be there in the (further) future" and tend to have "a narrative outlook on life" (Strawson 2004, p. 430). By contrast, Episodics are those who do not see their lives in narrative terms. For Episodics,

---

2   The period of 18 to 25 years of age is often called "emerging adulthood," which is a phase of the lifespan between adolescence and full-fledged adulthood that encompasses late adolescence and early adulthood, proposed by Jeffrey Arnett in 2000. See (Arnett 2000, pp. 469–80).

3   McAdams adopts a moderately reconstructive view of autobiographical recollections. Reconstruction involves the selection and interpretation of certain memories as self-defining memories. People grant privileged status to those self-defining memories in their narrative identities.

their identity states are rather discontinuous. Because their sense of self in any present moment bears no connection to their sense of self at any previous point in their history, their selves and lives are never organized coherently in narrative form (Eakin 2008, p. 11). While Strawson acknowledges that many people find themselves in a Diachronic (narrative) mode of self-experience, he insists that there are people who have an Episodic disposition (non-narrative), arguing that a "strongly Episodic life is one normal, non-pathological form of life for human beings, and indeed one good form of life for human beings, one way to flourish" (Strawson 2004, pp. 432–33). Strawson ultimately rejects the two narrativity theses: both the ethical and the psychological. In "Against Narrativity," Strawson specifically criticizes MacIntyre and others who endorse the ethical narrativity thesis by writing,

> It seems to me that MacIntyre, Taylor, and all other supporters of the ethical Narrativity thesis are really just talking about themselves. It may be that what they are saying is true for them, both psychologically and ethically. This may be the best ethical project that people like themselves can hope to engage in. But even if it is true for them it is not true for other types of ethical personality, and many are likely to be thrown right off their truth by being led to believe that Narrativity is necessary for a good life. (Strawson 2004, p. 437)

Strawson implies that Diachronics erroneously assumes that "there is something chilling, empty, and deficient about the Episodic life" (Strawson 2004, p. 431). They should not think that "the Episodic life is bound to be less vital or in some way less engaged, or less humane, or less humanly fulfilled" (Strawson 2004, p. 431). In other words, not every human being sees his or her life as a form of coherent narrative or story. Not every human being needs to tell stories in order to have a good and ethical life.

### 3.2. Living Episodically as the Unstoried Self

What does it mean to live an Episodic life? How do Episodics make sense of their lives and sense of "who they are"? By considering himself as "relatively Episodic," Strawson uses himself as an example:

> I have a past, like any human being, and I know perfectly well that I have a past. I have a respectable amount of factual knowledge about it and I also remember some of my past experiences 'from the inside', as philosophers say. And yet I have absolutely no sense of my life as narrative with form, or indeed as a narrative without form. Absolutely none. Nor do I have any great or specific interest in my past. Nor do I have a great deal of concern for my future. (Strawson 2004, p. 433)

He also describes his Episodic disposition in a slightly different way: "it seems clear to me, when I am experiencing or apprehending myself as a self, that the remoter past or future in question is not my past or future, although it is certainly the past or future of GS the human being... this is not a failure of feeling. It is, rather, a registration of a fact about what I am" (Strawson 2004, p. 433). As an Episodic human, Strawson appears to live in a series of present-tense moments. The past is undoubtedly alive, for him, only in the sense that it has shaped his present, "just as musicians' playing can incorporate and body forth their past practice without being mediated by any explicit memory of it" (Strawson 2004, p. 432). In essence, Strawson's identity "has no narrative, wants no narrative, and needs no narrative" (Long 2009, p. 12).

More recently (2015), Strawson wrote another provocative essay called, "I Am Not a Story," which focuses explicitly on questioning the psychological narrativity thesis, pioneered by such renowned psychologists as Jerome Bruner and McAdams (Strawson 2015). In response to Bruner's argument: "Self is a perpetually rewritten story," and McAdams' argument: "We are all storytellers, and we are the stories we tell," Strawson responds by saying, "I think it's false–false that everyone stories themselves, and false that it's always a good thing. These are not universal human truths . . . The narrativists are, at best, generalizing from their own case, in all-too-human way" (Strawson 2015). In this article, Strawson appears to question the basic Diachronic disposition in humanity, arguing that

"We're naturally–deeply–non-narrative" because the deliverance of memories is fundamentally "untrustworthy" and "hopelessly piecemeal and disordered," especially when we are trying to remember a temporally extended sequence of events (Strawson 2015, p. 2).

To support his argument, Strawson discusses the importance of human memories in the formation of narratives. McAdams explains that stories that are based on personal experiences begin with episodic memory. Episodic memory is "the ability to recall specific events (episodes) from the past" (McAdams 2013, p. 58). As the memory of the larger autobiographical data, it is a skill that enables humans "to travel backward in subjective time and to link remembered events to imagined future episodes" (McAdams, p. 58). Episodic memory provides the personal experience of time that people depend on when they construct and tell their stories. A considerable amount of what people remember may not be directly connected in their minds to specific events. Nonetheless, human beings are still able to travel back in time to recall specific episodes that have happened in their lives.

In response, Strawson uses the words of Michel de Montaigne to clarify his thoughts on autobiographical memory: "I can find hardly a trace of [memory] in myself . . . I doubt if there is any other memory in the world as grotesquely faulty as mine is!" (De Montaigne 1991, p. 32; Strawson 2015). Due to his poor ability to remember his past experiences, de Montaigne writes the unstoried life, the only present life that matters. Furthermore, because his memory is so untrustworthy, Montaigne concludes that the fundamental lesson of self-knowledge is that "self-knowledge comes best in bits and pieces" (Strawson 2015). Once again, while Strawson understands that there are many people who advocate and live out self-narratives, he points out that there are also people who have untrustworthy memories and a non-narrative outlook on life. Poor memory and an Episodic disposition are not necessarily hindrances for people to live a meaningful life. In regard to the psychological narrativity thesis, Strawson thinks that our past consciousness is frequently irrecoverable, which means human life cannot simply assume a story-like shape in the first place. Thus, as an "unstoried self," Strawson contends that narrative identity is not a universal human condition, and continues to insist that "what I care about . . . is how I am now" (Strawson 2004, p. 438).

## 4. The Case Study of the Lost Mariner's Religious Experience

While Strawson gives an example of what it is like to be an Episodic person by using his personal account, he is clearly generalizing as well from his own experience of discontinuous identity. In fact, he does not provide other examples of the Episodic mode of self-experience that are more empirical and descriptive. Then what is it like to be an Episodic person whose identity states are discontinuous? Can Diachronics and Episodics even understand each other? Are these two ways of self-experience radically opposed? What causes people to be more Diachronic or Episodic in their lives? As I consider these questions, I analyze the autobiographical writing of Leo Tolstoy presented in "The Varieties of Religious Experience" by William James. In "The Varieties", by examining Tolstoy's autobiographical writing, James presents Tolstoy as "a well-marked case of anhedonia, of passive loss of appetite for all life's values" (James [1902] 2002, p. 149). Best known for his novels "War and Peace" (1869) and "Anna Karenina" (1878), Leo Tolstoy (1828–1901) published his memoir, "A Confession" in 1882, which is a collection of his personal stories of midlife crises, as part of an existential struggle to find life's meaning.[4] James examines Tolstoy's personal account of existential crisis, where James diagnoses Tolstoy's condition as anhedonia, which refers one's inability to experience pleasure or joy. Anhedonia is considered to be a core feature of major depressive disorders and maladaptive behaviors (Gorwood 2008). Subsequently, James highlights Tolstoy's religious experience, which enabled Tolstoy to find a new sense of purpose in life as well as renewed happiness.

---

[4]  In 1887, another version of Tolstoy's memoir was published and titled, *My Confession and the Spirit of Christ's Teaching*.

At the age of fifty, Tolstoy begins to experience what James calls, "moments of perplexity" as if Tolstoy did not understand "how to live" or "what do to do" with his life (James [1902] 2002, p. 152). Tolstoy writes,

> I felt . . . that something had been broken within me on which my life had always rested, that I had nothing left to hold on to, and that morally my life stopped. An invincible force impelled me to get rid of my existence, in one way or another. It cannot be said exactly that I wished to kill myself, for the force which drew me away from life was fuller, more powerful, more general than any mere desire. It was a force like my old aspiration to live, only it impelled me in the opposite direction. It was an aspiration of my whole being to get out of life . . . I did not know what I wanted. I was afraid of life; I was driven to leave it; and in spite of that I still hoped something from it. (Tolstoy 1887; Quoted in James [1902] 2002, p. 153)

At this time, Tolstoy indicates externally that his life seemed successful with many professional achievements. Moreover, with loving and caring people around him, he "was more respected by [his] kinfolk and acquaintances that [he has] even been" and also writes, "I was loaded with praise by strangers; and without exaggeration I could believe my name already famous" (Tolstoy 1887; Quoted in James [1902] 2002, p. 154). But, at the same time, Tolstoy confesses how inner self was being tormented and devastated by "despair—the meaningless absurdity of life" (Tolstoy 1887; Quoted in James [1902] 2002, p. 154). Tolstoy writes, " . . . I could give no reasonable meaning to any actions of my life. And I was surprised that I had not understood this from the beginning. My state of mind was as if some wicked and stupid jest was being played upon me by someone" (Tolstoy 1887; Quoted in James [1902] 2002, p. 153). Here, Tolstoy demonstrates his highly "Episodic" disposition, raising a series of unanswered questions: "What will be the outcome of what I do to-day? Of what I shall do to-morrow? What will be the outcome of all my life? Why should I live? What should I do anything?" (Tolstoy 1887; Quoted in James [1902] 2002, p. 155). In other words, he finds himself stuck in present limbo, having no ability to make sense of his achievements in the past as well as to imagine and envision his future. In such deep despair, Tolstoy writes that he constantly struggles with "hiding the rope in order not to hang myself from to the rafters of the room where every night I went to sleep alone" (Tolstoy 1887; Quoted in James [1902] 2002, p. 153).

However, after three years of "unending questioning" with a sense of inner discordance, Tolstoy finally experiences an intense divine encounter that changes everything. He writes in vivid detail about the day when he had a personal religious experience with vivid details:

> I remember one day in early spring, I was alone in the forest, lending my ear to its mysterious noises. I listened, and my thought went back to what for these three years it always was busy with—the quest for God. But the idea of him, I said, how did I ever come by the idea? And again there arose in me, with this thought, glad aspirations towards life. Everything in me awoke and received a meaning. (Tolstoy 1887, pp. 64, 65; Quoted in James [1902] 2002, p. 185)

For Tolstoy, it is this instant, yet powerful divine encounter with God that brought back clarity and a sense of self in him, which ultimately saved him from committing suicide: "After this, things cleared up within me and about me better than ever, and the light has never wholly died away . . . as gradually and imperceptibly did the energy of life come back." (Tolstoy 1887; Quoted in James [1902] 2002, p. 185). For many years, Tolstoy tries to find answers to the unending questions by seeking "the life of conventionality, artificiality, and personal ambition" and living "the life of upper, intellectual, artistic classes" (James [1902] 2002, pp. 184, 185). But, his encounter with God taught Tolstoy that he "had no right to rely on [his] individual reasoning" and the questions can be only answered by God (James [1902] 2002, p. 184). With this newly gained aspiration, Tolstoy eventually

realizes that while the three years of anhedonia with suicidal melancholia was excruciating, it was a meaningful "quest of God." For Tolstoy, God is now the main source of purpose and happiness in life, giving him reasons to live a life. By listening to "mysterious noises" from God and embracing them fully, Tolstoy experiences a divine encounter, which brings him a sense of continuity in experiencing clarity, purpose, and peace in his inner life. In James' view, this divine experience for Tolstoy was to reorganize "his soul in order, the discovery of its genuine habitat and vocation, the escape from falsehoods into what for him were ways of truth. It was a case of heterogeneous personality tardily and slowly finding its unity and level" (James [1902] 2002, p. 186). Through this particular religious experience, Tolstoy, in the end, comes to interpret the three grueling years of his inner struggle as "a quest" to encounter God intimately and indicates that he has found now a renewed sense of purpose and meaning in life with "the positive willingness to live" (James [1902] 2002, p. 187).

## 5. The Episodic Becoming Diachronic Through Religious Experience

James' case study of Tolstoy, namely his religious experience, is an empirical account that demonstrates how an individual like Tolstoy who has an Episodic disposition due to anhedonia can possibly gain a Diachronic disposition by having a particular religious experience. For a long time, Tolstoy struggles with despair and suicidal melancholia, which causes him to experience "the meaningless absurdity of life" (Tolstoy 1887; Quoted in James [1902] 2002, p. 154). He does not find his past valuable and is unable to envision his bright future. But in the end, through his religious experience, Tolstoy is now able to find purpose, clarity, value, and most importantly, his sense of self. Tolstoy's Episodic world finds a level of coherence, continuity, and clarity by having a personal divine connection. Tolstoy comes to gain a sense of "Diachronic" narrative identity.

How does religious experience enable and shape formation of one's identity? Religious experience, according to James, includes "the feelings, acts, and experiences of individual men in their solitude, so far as they apprehend themselves to stand in relation to whatever they may consider the divine" (James [1902] 2002, p. 31). In "The Varieties of Religious Experience" (originally published in 1902), James explores how one's religious self is created and developed by one's religious experience. By investigating the personal accounts of numerous religious individuals, James shows how these individuals come to their sense of self as they encounter a variety of religious experiences, such as conversion, spiritual enlightenment, attainment of saintly characteristics, and religious practices (Cho 2019). By having religious experience or being religious, people create and "establish [themselves] in possession of ultimate reality" (James [1902] 2002, pp. 500, 501). For example, James explains that prior to having religious experience, many individuals frequently struggle with "a certain discordancy or heterogeneity" in their sense of self, often with "an incompletely unified moral and intellectual constitution" (James [1902] 2002, p. 167). But as they encounter religious experience, their sense of discordancy or heterogeneity "emerged into the smooth waters of inner unity and peace" (James [1902] 2002, p. 175). In the same way, the "discordancy or heterogeneity" in Tolstoy's Episodic character of identity became more "unified" and "concordant" while Tolstoy's "feelings, acts, and experiences" in relation to what he considers "divine" were embodied, practiced, written and told in the chapel. James further elaborates this process:

> To be converted, to be regenerated, to receive grace, to experience religion, to gain an assurance, are so many phrases which denote the process, gradual or sudden, by which a self hitherto divided, and consciously wrong, inferior, and unhappy, becomes unified and consciously right, superior and happy, in consequence of its firmer hold upon religious realities. (James [1902] 2002, p. 189)

As a psychological process, James claims that "religion is only one out of many ways of reaching unity" and "the process of remedying inner incompleteness and reducing inner discord" in human lives (James [1902] 2002, p. 175). In other words, religious

experience enables "the process of unification" within the Episodic state of human identity and eventually creates a sense of coherence and continuity.

Consequently, James describes religious experience as a psychological process that "religious ideas, previously peripheral in his consciousness, now take a central place, and that religious aims form the habitual centre of his energy" (James [1902] 2002, p. 196). Through such religious experiences, one eventually finds, develops, and solidifies one's religious self that is "the very core" of the self. Prior to his publication of "The Varieties", in "The Principles of Psychology" (originally published in 1890) and later in "Psychology: The Briefer Course" (originally published in 1892), James discusses the essence of the religious self. The religious self or the spiritual self is "the very core" of our self, composed of intellectual, moral, and religious aspirations and conscientiousness (James [1878] 1992, p. 178). The spiritual self, the "nucleus of our self, as we know it, the very sanctuary of our life, is the sense of activity which certain inner states possess. The sense of activity is often held to be a direct revelation of the living substance of our Soul" (James [1878] 1992, p. 178). Because the religious self is our most subjective and intimate self, this particular self is more concrete and permanent than any of our other selves (James [1878] 1992, p. 178). Aspects of the religious identity include elements such as personality, core values, ideology, beliefs, morality, and conscience that do not typically change throughout people's lifetimes. James argues that formation and understanding of the religious self is more rewarding than satisfying the needs of the social and material selves.[5] That is why James places the religious self at the top of the hierarchy of the empirical selves by highlighting the hierarchical scale as "the bodily me at the bottom, the spiritual me at top, and the extra-corporeal material selves and the various social selves between" (James [1878] 1992, p. 186). The religious identity, in other words, is "so supremely precious that, rather than lose it, a man ought to be willing to give up friends and good fame, and property, and life itself" (James [1878] 1992, p. 187). The essence and strength of the religious identity explains how Tolstoy was able to gain a sense of his identity with "continuity, clarity, and unity" while experiencing an intimate divine encounter because his discordant state of being became "unified" within "the very core" of himself.

## 6. Conclusions: We May Be Blends of Both

As for Strawson's anti-narrative argument, many, including myself, would disagree with him about the validity of the narrativity theses. But Strawson makes a valuable point, that people with the Episodic disposition exist and they certainly do not see themselves as narratives or stories. It also challenges scholars in various fields who employ narrative approaches to reassess their use of narrative in their scholarship. As an extreme Episodic, Strawson asserts that his is not a storied self and has no interest in emplotting his life into a meaningful story. He does not believe that he needs a coherent story in order to live a good and virtuous life. He has never had in the past, and will never have in the future, a narrative outlook on life, emphasizing the clear distinction between the two ways of being. But once again, Strawson is evidently generalizing from his own point of view. Then how can we appropriately sort people into Strawson's two categories?

For example, if I am a Diachronic and someone asks me to tell about myself or my story, I know I will eventually get around to telling the story of my life. But if I were to tell my story honestly and coherently, I would need hours, days, and even years to reflect on my past, internalize disconnected episodes, and finally construct a meaningful plot line. Similarly, if you ask people whether they believe in continuous identity, many would say they do. If you ask them, though, about the extent to which they can actually remember and evoke their past, pressing them as to whether they can re-experience in the present their earlier states of consciousness, many of these previously unreflecting Diachronics

---

5   In *The Principles of Psychology*, James divides the Me or empirical self into three components: (1) the material self—including the physical body and all possessions intimately associated with it; (b) the social self—including the recognition a person receives from his or her significant relationships; and (c) the spiritual self—including the entire collection of one's state of consciousness (See James [1890] 1950, pp. 291, 292).

would find themselves being fairly Episodic as well (Eakin 2008, p. 13). Likewise, even convinced Episodic individuals now and then could see "a glimpse of some larger narrative into which, willingly or unwillingly, they have been drawn and which arranges however loosely, the discrete episodes of their lives" (Long 2009). Therefore, contrary to Strawson's argument, human beings may find themselves in both Episodic and Diachronic camps in various seasons of life.

The case of Tolstoy also suitably demonstrates that he was both Episodic and Diachronic. Since the age of fifty, Tolstoy lived with a highly Episodic disposition due to his suicidal melancholia, but after his vibrant religious experience, Tolstoy became more Diachronic. As he encountered God intimately and gained a renewed sense of purpose in his life, Tolstoy then had new possibilities to imagine and he developed a narrative outlook on life and constructed a continuous sense of self, a religious self in particular. In addition, Virginia Woolf and Henry James, both writers whom Strawson cites as Episodics, actually did make a narrative turn at the end of their careers (Eakin 2008, p. 13). These primarily Episodic writers eventually turned autobiographers by revisiting and recovering crucial parts of their earlier selves. They also became stories. That is to say that Episodics do not have to live in an "episodic" or "discontinuous" state permanently. Episodics can be open to gain an interest in seeing their lives as meaningful narratives. Religion, certainly, can facilitate and enable Episodic people to make this narrative turn through a variety of religious experience. In the same way, Diachronics need to be reminded that their life stories do not have to be coherently unified all the time.

Richard Kearney, philosopher and novelist, explains that narrative and its "notion of continuous experience, associated with traditional linear narrative, has been fundamentally challenged" due to the emergence of "hyper-advanced telecommunications and digital data flows" and also the face of violence, abuse, illness, and trauma in the human experience (Kearney 2001, pp. 125, 126). The limits and difficulties of narrative cannot be denied. Nonetheless, stories still offer us some of the richest and most enduring insights into human identity and the sociocultural contexts in which we live. "Storytelling will never end," Kearney argues, because telling a story always "invites us to become not just agents of our own lives, but narrators and readers as well. It shows us that the untold life is not worth living (Kearney 2001, p. 156). There will be always somebody to say, "tell me a story," and someone there to respond by saying, "Once upon a time . . . " (Kearney 2001, p. 126).

**Funding:** The research received no external funding.

**Conflicts of Interest:** The author declares no conflict of interest.

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
