# Peer review of "Do We All Live Story-Shaped Lives? Narrative Identity, Episodic Life, and Religious Experience"

_religions, doi:10.3390/rel12020071_

Round 1
Reviewer 1 Report
This is an excellent comparative analysis of the narrative and episodic approaches to personal identity. The author is well-versed in the literature on the subject and demonstrates competent handling of the concepts as well as of the strengths and weaknesses of the two theories. The thesis is very intriguing, although I have some doubts about the author's claims regarding the conclusions that can be safely drawn from Oliver Sacks' observations of patient Jimmy's participation in practices of religious worship. The author argues that, as a result of partaking in religious practice, Jimmy "became a story." (p. 11, line 469). The evidence for this conclusion based on the report of Sacks' observations is very thin. Simply because Jimmy's attention and concentration competencies were improved significantly does not entail that Jimmy developed a narrative sense of self. It strikes me as very speculative to conclude what, if any, sense of narrative unity Jimmy might have achieved on the basis of the observation that he had become more attentive, pensive and peaceful. (p. 8, line 363) I do agree with the author's claim that religious experience can bring about narrative unity out of a fragmented episodic self (that's basically an elaboration of William James' thesis that I find very plausible). But I'd like to see much more evidence that this kind of unification indeed had taken place in Jimmy's life. That's why I feel unable to judge the scientific soundness of the study.
A couple of small edits:
p. 1, line 36: add "as" in "to experience their lives [as] a narrative or story..."
p. 1, lines 36-37: a qualifier is missing the question: "How are non-narrative and narrative modes [different?]"
p. 2, line 53: "theses contained [in] the concept..."
p. 2, line 66: Italicize After Virtue
p. 5, line 215: "In his argument against [the] two narrative identity theses..."
p. 6, line 249: This is grammatically incorrect: "Not every human being needs to be stories..."
p. 6, line 256: change "and" before "philosophers say" to "as philosophers say"
p. 6, line 266: "and body forth their practice" - what's "body forth"?
p. 6, line 274: change "universal" to "universally"
p. 7, line 293: delete the superfluous "that"
p. 8, line 366: change "religion" to "religious"
p. 9, line 410: change "conscious" at the end of the line to "consciously"
p. 9, line 428: add period and citation in parentheses after the quotation ending with "our Soul"
p. 10, line 437: Is the word in the quotation from James "purport" or "proport"?
Author Response
I deeply appreciate the reviewer's careful review of my manuscript. It was very helpful to read what the reviewer considers the strength and weaknesses of the manuscript. I do understand and agree with the reviewer's main critique that Sack's observation of Jimmie is not strong enough to prove and support the main argument that Episodics can possibly become Diachronic through religious experience. So I made a decision to take out Sack's case study and instead add a different case study that William James presents in The Varieties of Religious Experience. James presents autobiographical writing of Leo Tolstoy who suffers from depression and suicidal melancholia, but experiences renewal through a personal religious experience. This case study highlights Tolstoy's personal experience (in his own words) for the readers to observe and understand his Episodic and Diachronic dispositions as well as the significance of his divine encounter with God. I think this is a stronger case study than Sack's for the readers to understand that this kind of unification indeed can take place in one's identity through a personal religious experience.
Reviewer 2 Report
Please see attached file

Author Response
I deeply appreciate the reviewer's careful review of my manuscript. It was very helpful to read what the reviewer considers the strength and weaknesses of the manuscript. The reviewer's thorough comments guided me to revisit and revise the manuscript more critically.
1) I do understand and agree with the reviewer's main critique that Sack's observation of Jimmie is not strong enough to prove and support the main argument that Episodics can possibly become Diachronic through religious experience. So I made a decision to take out Sack's case study and instead add a different case study that William James presents in The Varieties of Religious Experience. James presents autobiographical writing of Leo Tolstoy who suffers from depression and suicidal melancholia, but experiences renewal through a personal religious experience. This case study highlights Tolstoy's personal experience (in his own words) for the readers to observe and understand both of his Episodic and Diachronic dispositions as well as the significance of his divine encounter with God. I think this is a stronger case study than Sack's for the readers to understand that this kind of unification indeed can take place in one's identity through a personal religious experience.
2) I mainly analyze MacIntyre and McAdams because these two are among the narrativist scholars that Strawson argues against in his anti-narrativist discourses. I thought it would be more helpful for the readers to know more about the main ideas of MacIntyre and McAdams because Strawson does not provide details of their ideas. I added a footnote explaining why I am using MacIntyre and McAdams in particular for the purpose of this paper.
3) I looked through Strawson's article and saw that while he does not give his specific definition of what "narrativity" is, he helps us to know that narrativity is what helps people to narrativize their stories and identities. But I thought providing another more concrete definition of "narrativity" would be helpful for the readers.
Round 2
Reviewer 2 Report
The revised version of this paper is much improved and addressed my concerns and comments regarding the original submission.
The decision by the author(s) to replace Jimmie's narrative with Tolstoy's avoids many of the issues I previously identified, and provides a far stronger basis on which to make the argument about episodicity and diachronicity.
For me, the key contribution this articles makes is the argument that episodicity and diachronicity are not necessarily mutually exclusive positions in an individual's life but that individuals may at different points experience their lives in one manner or the other. I would suggest that the author(s) include this conclusion in the abstract so as to notify potential readers that this is the originality of the piece.
I still think that lives 51-54 on the definition of narrativity imply this is the definition with which Strawson is working. I would suggest that these lines be reworded, perhaps along the following lines: "Strawson first focuses on the concept of narrativity as helping people to narrativise their stories and identities, but does not provide a specific definition. A clearer definition of narrativity, and one which aligns well with Strawson's implied position, is found in Christman (2004, p.696): "an identifiable .........".
The paragraph on Tessman (lines 118-137) is interesting but I am unclear as to how this fits with the rest of the piece. Nothing seems to be taken from this discussion into the later discussion of Tolstoy's narrative or the concluding arguments of the author(s). I suggest either referring back to the points Tessman makes about oppression later on with reference to Tolstoy's narrative, or shortening or omitting this paragraph altogether.
There is a slight issue of voice in lines 148 following. The first paragraph starts with 'we', the second with 'you', and the second paragraph then returns to 'we'.
In the last paragraph on p8 (lines 366ff) I think there is a slight elision of diachronicity with meaning and episodicity with meaninglessness. In locating himself within the larger narrative of religion Tolstoy found meaning - it is unclear that this was a movement from episodicity rather than simply a movement from meaninglessness. A sentence or two addressing this would be helpful for the reader.
I would like to see a reference to support the claim on lines 436-7 that "Because the religious self is our most subjective and intimate self, this particular self is more concrete and permanent than any other selves". If this is James give a page reference. If this is the author(s)' comment then a supporting reference is required.
There is a typo on line 514 - it should read 'A Tale of Two Stories'.
Author Response
Thank you so much for another round of careful peer review. The feedback and comments were very helpful.
1) In the abstract, I added how the key contribution of this article is to show that episodicity and diachronocity are not always mutually exclusive and people find themselves in both at different points in life.
2) I addressed how Strawson does not give a specific definition of "narrativity," and I decide to use John Christman's definition, which is aligned with Strawson's understanding.
3) I do understand that Tessman might not need to be in the article. But I mention Tessman to point out that MacIntyre's idea is not without limitation. So I decided to shorten this paragraph.
4) I do appreciate your point on how there's an elision of diachronicity with meaning and episodicity with meanglessness. To make it more clear, I quote Tolstoy's words to show that for Tolstoy, his experience of melancholia was to experience "the absurdity of meaningless."
5) I made all other changes according to the reviewer's suggestions.